# Flavonoid-Rich Fractions of *Bauhinia holophylla* Leaves Inhibit *Candida albicans* Biofilm Formation and Hyphae Growth

**DOI:** 10.3390/plants11141796

**Published:** 2022-07-07

**Authors:** Sara Thamires Dias da Fonseca, Thaiz Rodrigues Teixeira, Jaqueline Maria Siqueira Ferreira, Luciana Alves Rodrigues dos Santos Lima, Walter Luyten, Ana Hortência Fonsêca Castro

**Affiliations:** 1Laboratory of Natural Products, Postgraduate Program in Biotechnology, Campus Centro-Oeste, Universidade Federal de São João del-Rei, Divinópolis 35501-296, MG, Brazil; sara.tdiasf@gmail.com (S.T.D.d.F.); jaque@ufsj.edu.br (J.M.S.F.); luarsantos@ufsj.edu.br (L.A.R.d.S.L.); 2Department of Biomolecular Sciences, School of Pharmaceutical Sciences of Ribeirão Preto, Universidade de São Paulo, Ribeirão Preto 14040-900, SP, Brazil; thaiz_rt@hotmail.com; 3Department of Biology, Faculty of Science, Katholieke Universiteit Leuven, 3000 Leuven, Belgium; walter.luyten@kuleuven.be

**Keywords:** *Bauhinia*, antibiofilm, flavonol-3-*O*-glycosides, medicinal plant, yeast–hyphae transition

## Abstract

This study evaluated the effect of the extract and fractions of *Bauhinia holophylla* on *Candida albicans* planktonic growth, biofilm formation, mature biofilm, and hyphae growth. Three *C. albicans* strains (SC5314, ATCC 18804, and ATCC 10231) were tested. The crude extract and the fractions were obtained by exhaustive percolation and liquid–liquid partition, respectively. Phytochemical analyses of *B. holophylla* extract and fractions were performed using high-performance liquid chromatography coupled with a diode-array detector and mass spectrometry (HPLC-DAD-MS). A microdilution assay was used to evaluate the effect of the *B. holophylla* extract and fractions on *C. albicans* planktonic growth, and crystal violet staining was used to measure the total biomass of the biofilm. Hyphae growth was analyzed using light microscopy. Thirteen flavonoids were identified, with a predominance of the flavonol-3-*O*-glycoside type based on quercetin, myricetin, and kaempferol. Flavonoid-rich fractions of *B. holophylla* leaves displayed antifungal activity and inhibited both biofilm formation and hyphae growth in all the tested strains, but were not effective on *C. albicans* planktonic growth and mature biofilm. This study indicates that flavonoid-rich fractions from *B. holophylla* leaves interfere with the virulence of *Candida* species and support the use of *Bauhinia* spp. in folk medicine to treat infections.

## 1. Introduction

*Candida albicans* is the major opportunistic pathogen of invasive fungal infections in humans [1,2]. *Candida* spp. infections represent 90% of invasive infections in hospitalized cases, mainly in patients undergoing intensive care, which can lead to death in up to 80% of cases [3,4]. Expression of adhesins and invasins on the cell surface, thigmotropism, the secretion of hydrolytic enzymes, the transition between yeast and hyphal forms, and biofilm formation are the main virulence factors of *C. albicans* [5]. The morphological transition between yeast and hyphal forms is an indicator of virulence because while the yeast form is identified as a disseminated form, the hyphal form is invasive to tissues and target cells and more resistant to phagocytosis, due to its ability to invade and kill macrophages [6,7]. 

In addition to the dimorphic (yeast–hyphae) transition, biofilm formation also increases virulence and fungal resistance by allowing *Candida* to grow and colonize inert surfaces, such as implants and urinary catheters, or host tissues, causing effects on human health [8,9]. Fungal biofilms are aggregates of microorganisms immobilized in a matrix containing extracellular polymeric substances of microbial origin [10,11]. Its formation occurs in four stages: (1) initial adhesion, (2) cell proliferation and the initial phase of filamentation, (3) biofilm maturation, and (4) release of yeast cells [12,13]. *Candida* cells found in a biofilm are more resistant to antimicrobials, UV radiation, dehydration, and disinfection, due to the complex three-dimensional architecture of the biofilm, the impermeable matrix, upregulation of efflux pumps, and metabolic plasticity [10,13,14]. Along with virulence factors, the emergence of strains resistant to available antifungal agents has increased due to new resistance mechanisms, and until now, there has been no specific antibiofilm drug [12]. Therefore, the treatment of fungal infections has become more difficult and burdens health systems across the world, requiring a search for new treatment alternatives. 

Flavonoids are one of the major groups of bioactive compounds found in the *Bauhinia* genus. They correspond to an important class of plant-derived secondary metabolites, with demonstrated effects against bacteria, fungi, and viruses [15,16,17,18]. Several studies have reported the antibiofilm effects of flavonoids and their derivatives on bacteria and fungi. They are potential inhibitors of biofilm formation due to their ability to reduce the extracellular matrix [19,20], and they also inhibit the aggregation and maturation of the biofilm by decreasing surface hydrophobicity [21,22]. 

*Bauhinia holophylla* is a native species from the Brazilian Cerrado originally used in folk medicine to treat diabetes [23,24]. The hypoglycemic effects of the crude extracts of leaves were studied by Pinheiro et al. [24], Camaforte et al. [25], and Saldanha et al. [26]; in addition, Rozza et al. [27] observed their anti-ulcerogenic effects. Antiviral effects were also described by Santos et al. [28] against the dengue virus. Recently, Ribeiro et al. [29] and Marena et al. [30] reported the absence of cytotoxic and mutagenic effects, and even protective activity against known carcinogens, as well as antimicrobial effects of hydroalcoholic leaf extract, suggesting that *B. holophylla* has potential as a herbal medicine. 

Currently, there are no reports on the antibiofilm activity of *B. holophylla*. Therefore, considering the presence of flavonoids in this species and the traditional use of some *Bauhinia* species to treat infections [31], this study aimed to investigate the antibiofilm activity of extracts and fractions of *B. holophylla* leaves against *Candida albicans* biofilm and hyphae growth.

## 2. Results

### 2.1. Phenolic Compounds’ Content

The total phenolic compounds’ contents in crude extract and fractions of *B. holophylla* leaves are shown in Table 1. The levels of total phenols and flavonoids ranged from 20.46 to 47.45 µg GAEq mg^−1^ DS and 2.37 to 9.93 µg QEq mg^−1^ DS, respectively. A higher total flavonoid content was observed in the DCM fraction, followed by the EtOAc fraction (*p* < 0.05).

### 2.2. Chemical Composition of the Crude Extract and Fractions of B. holophylla

The samples of *B. holophylla* were analyzed using HPLC-DAD-MS to identify their chemical constituents (Figure 1, Table 2).

The patterns of fragmentation of the reference compounds reported in the literature, compared to those obtained in the present study, allowed us to identify 13 flavonoids, with predominance of the flavonol-3-*O*-glycoside type based on quercetin, myricetin, and kaempferol. The chromatogram peaks with absorption bands at 240-280 and 330-350 nm are typical of flavonoid derivatives of flavonols [32]. Mass fragments at *m/z* 303 and 319 characterized quercetin and myricetin, respectively. The ions at *m/z* [M+H]^+^ 435, 449, 465, and 479 were dereplicated as quercetin-*O*-pentoside, quercetin-*O*-deoxyhexoside, myricetin-*O*-hexoside, and myricetin-*O*-hexoside, respectively, similar to the results obtained by Camaforte et al. [25] and Rozza et al. [27]. The chemical profiles of the fractions are similar to that of the crude extract (CHE), except for the hydroethanolic (EtOH) fraction.

### 2.3. Evaluation of the Planktonic and Antibiofilm Activity against Candida albicans of Extract and Fractions

The planktonic growth of *C. albicans* strains was not altered when treated with the *B. holophylla* extract and fractions for 48 h (Table 3). However, *C. albicans* strains were susceptible to the positive controls, as expected. *C. albicans* SC5314 and *C. albicans* ATCC 18804 growth decreased significantly due to fluconazole (MIC= 1.95 µg/mL) and. *C. albicans* ATCC 10231 due to nystatin (MIC= 3.91 µg/mL). 

### 2.4. B. holophylla Fractions Inhibit Biofilm Formation of C. albicans Strains, but Do Not Have Effect on Mature Biofilm 

*C. albicans* biofilm formation was not inhibited by *B. holophylla* extract, but the EtOAc, DCM, and EtOH fractions were effective in all of the tested strains. The three fractions tested decreased *C. albicans* SC5314 and *C. albicans* ATCC 18804 biofilms, and the EtOAc and EtOH fractions showed inhibitory effects against *C. albicans* ATCC 10231 (*p* < 0.05). Biofilm formation by *C. albicans* strains was dramatically inhibited by nystatin, the positive control (Figure 2).

*C. albicans* SC5314 biofilm formation decreased significantly with the EtOAc fraction with a highly inhibitory effect from 1250 to 62.5 μg/mL (82 to 72% inhibition) (Figure 2A). An inhibitory effect similar to the nystatin (about 90%) was observed at a concentration of 125 μg/mL, and from 31.25 μg/mL, the inhibitory effect dropped to 18% and was progressively reduced further in the other concentrations (*p* < 0.05). *C. albicans* SC5314 biofilm formation was also affected by DCM fraction, especially at 1250 μg/mL with 79% inhibition and an EtOH fraction at 1250 to 625 μg/mL (60 and 55% inhibition, respectively). *B. holophylla* fractions significantly inhibited *C. albicans* ATCC 18804 biofilm formation (*p* < 0.05) (Figure 2B). Inhibition was greater than 60% at 1250 to 625 μg/mL for all the fractions. EtOAc and DCM maintained this inhibition at 312.5 and 125 μg/mL, respectively. From 312.5 μg/mL, a progressive reduction in the inhibitory effect of the EtOH fraction was observed. *C. albicans* ATCC 10231 biofilm formation decreased significantly in a concentration-dependent manner with the EtOAc fraction (*p* < 0.05) (Figure 2C). The maximum inhibitory effect (75%) was reached when it was treated with 1250 and 625 μg/mL of the EtOAc fraction. From 312.5 to 7.8 μg/mL, the inhibitory effect decreased to about 35 to 20%, and from 3.9 to 0.25 μg/mL, the inhibition was approximately 10% on average.

This result was less prominent for the EtOH fraction, which showed some inhibition at concentrations from 625 to 125 µg/mL, with 27% inhibition of biofilm formation at 125 µg/mL.

Results show a potential inhibition of *C. albicans* biofilm formation by *B. holophylla* fractions, especially EtOAc, which presented a marked inhibition for all three strains tested (Figure 3). This effect was more pronounced on the SC5314 strain, where the inhibition of biofilm formation was greater than 80% at concentrations between 1250 and 312.5 μg/mL, and remained above 70% for concentrations from 125 to 62.5 μg/mL.

Although the fractions had a pronounced inhibitory effect on biofilm formation, extract and fractions of *B. holophylla* did not have an effect on mature the biofilm of *C. albicans* strains (*p* > 0.05) (Figure 4). 

### 2.5. B. holophylla Fractions Reduce Yeast–Hyphae Transition

The fractions of *B. holophylla* were able to reduce the yeast–hyphae transition at 625 µg/mL, compared to untreated cells, depending on the studied strain (Figure 5). These effects could be observed mainly after 24 and 48 h of incubation. The *B. holophylla* DCM fraction reduced the yeast–hyphae transition of *C. albicans* SC5314, since growth was reduced at 24 and 48 h (Figure 5A–I). Treatment of *C. albicans* ATCC 18804 with the *B. holophylla* EtOAc fraction markedly reduced the yeast–hyphae transition at 24, 48, and 72 h, while the DCM fraction was active mainly in the first 24 h (Figure 5J–R). For *C. albicans* ATCC 10231, the EtOH fraction led to the lowest number of cells growing in the hyphal form after 24, 48, and 72 h of incubation (Figure 5S–X). The yeast–hyphae transition of *C. albicans* strains was dramatically inhibited by nystatin, the positive control.

## 3. Discussion

The therapeutic effects of several medicinal plants are attributed to the abundant presence of flavonoids as bioactive compounds [33]. Flavonoids are ubiquitous natural products found in the plant kingdom, often in a glycoside form. Flavonols correspond to one of the main classes of flavonoids, being chemically a 3-hydroxy derivative of flavones [9,33]. Their wide distribution in nature, low cost, and low toxicity are advantages that make flavonoids good candidates for application in antifungal therapy [34].

In the current study, we observed that fractions of *B. holophylla* leaves containing, predominantly, flavonoids of the flavonol-3-*O*-glycoside type were able to inhibit biofilm formation of *C. albicans* strains and the yeast–hyphae transition in vitro. However, these fractions showed no effect on *C. albicans* planktonic growth and mature biofilms. The ethyl acetate fractions, containing flavonol-3-*O*-glycosides based on quercetin, myricetin, and kaempferol, exhibited a marked activity against biofilm formation on all strains tested, but especially on the SC5314 strain. Similarly, these fractions were able to reduce the yeast–hyphae transition at 625 µg/mL, depending on the studied strain.

*Bauhinia* spp. are traditionally used in folk medicine to treat infections, pain, inflammations, and diabetes [31]. Several of these activities have been attributed to flavonoids and scientifically proven by in vivo and in vitro models [25,28,35]. Currently, there are few studies on the antimicrobial effects of *B. holophylla*. Marena et al. [30] reported antibacterial and antifungal effects against some microorganisms, but these activities have not been confirmed by Fonseca et al. [36]. The reason for this discrepancy is not clear. 

Bioguided studies of extracts and fractions of *B. holophylla* leaves have led to the identification of several glycosylated flavonoids and aglycones [25,27]. There are complex glycosylated flavonoids with up to five sugar residues, and some studies have shown that different glycosylated flavonoids exert different biological activities [37]. Flavonoids can be synthesized in plants in response to microbial infection. However, the mechanisms of antifungal activity and the antivirulence properties of flavonoids remain unknown [33,38,39]. 

*B. holophylla* extract and fractions did not inhibit *C. albicans* growth, but they were able to inhibit biofilm formation and also reduced the yeast–hyphae transition. Most *C. albicans* infections are associated with biofilm formation, an important virulence factor, since the biofilm is resistant to most available antifungal drugs [40]. In addition, the dimorphic transition from a budding yeast cell to a filamentous form (yeast–hyphae) also represents another important virulence factor, since the hyphae contribute to the stability of biofilms, and their ability to adhere is critical for tissue penetration [12,13,41]. These factors not only facilitate adherence to and penetration into the host tissue, but they also evade host immunity [42]. These two virulence attributes are primary sources of multiple-drug-resistance development and invasive infections, because they are difficult or even impossible to eradicate with conventional antifungal agents [43,44]. In this way, the flavonoid-rich fractions of *B. holophylla* leaves are potential candidates for use in combination with established antifungal drugs to treat *C. albicans* infections, since they may act in different stages of biofilm formation. However, some studies showed that antibiofilm activity can vary among *Bauhinia* spp., since flavonoids from a *B. forficata* extract exhibited strong antifungal activity at 15.62 µg/mL on the planktonic form, but lower inhibition on biofilm formation (45%) [45]. Therefore, preventing biofilm and hyphae formation could facilitate the treatment of *C. albicans* infections, because the number of available anticandidal drugs classes is limited [46].

Flavonoids of the flavonol-3-*O*-glycoside type are an important group of chemical compounds that also have a broad-spectrum antifungal activity. Quercetin, myricetin, kaempferol, and their glycosylated forms have been reported as therapeutic molecules for human health, and emerge as promising novel antifungals [47,48,49]. Despite their antifungal activity, some reports demonstrate that when applied alone, quercetin, e.g., exhibits low antifungal activity, but in combined therapy with fluconazole, there is a strong synergism in the clinical management of *C. albicans* biofilms [44,50]. Just like quercetin, myricetin also has multiple biological actions [51], and some studies showed that myricetin has antifungal and potent antibiofilm activities, therefore being able to enhance the antifungal effect of miconazole in combined therapy [52]. Quercetin and myricetin also have potential to control hyphae growth [53]. In addition, kaempferol, an active flavonoid, has been considered a potential candidate drug against *Candida* species, and the concomitant use of fluconazole and kaempferol demonstrated a kaempferol-induced reversion in fluconazole-resistant *C. albicans* [46]. 

## 4. Materials and Methods

### 4.1. Chemicals

The following chemicals were purchased from different manufacturers: nystatin (Pharma Nostra, Rio de Janeiro, Brazil), fluconazole (Fagron, São Paulo, Brazil), ethanol (AlphaTec^®^, Santo André, Brazil), hexane, dichloromethane, and ethyl acetate (Cromato Produtos Químicos^®^, Diadema, Brazil), Sabouraud Dextrose Broth (Acumedia, San Bernardino, CA, USA), and Fetal Bovine Serum (Gibco^®^, São Paulo, Brazil). Dimethylsulphoxide (DMSO), crystal violet, RPMI 1640 (Roswell Park Memorial Institute Medium, Buffalo, New York), and MOPS (3-[N-Morpholino] propane sulfonic acid) were purchased from Sigma-Aldrich Co. (St. Louis, MI, USA). RPMI1640 medium (without NaHCO_3_ and with L-glutamine) was buffered with 0.165 M of MOPS to pH 7.4.

### 4.2. Plant Material and Preparation of the Extract and Fractions

*Bauhinia holophylla* (Bong.) Steud. (Fabaceae: Cercideae) leaves were collected in the Brazilian Cerrado in Ijaci, Southern Minas Gerais State, Brazil (21°09′97″ S and 44°55′65″ W GRW, at 835 m altitude) (SISBIO n° 24542-3, IBAMA Registration: 5042260). Fertile samples were collected, and the vouchers were identified by Andreia Fonseca Silva of the PAMG Herbarium (PAMG 57021) at the Agricultural Research Company of Minas Gerais (EPAMIG). This work has access permission for plant genetic heritage components (No. 010500/2014-6/CNPq/CGEN/MMA), and is registered on the SisGen Platform (Register A12A940), according to the Brazilian Biodiversity Law (13.123/2015). 

The plant material was dried in a ventilated oven (TE-394/500L, Tecnal; Piracicaba, Brazil) at 40 °C for 24 h, and pulverized in a knife mill (SL-31, Solab; Piracicaba, Brazil). The dried and powdered leaves (400 g) were extracted by exhaustive percolation using 70% ethanol as the extraction solvent for 7 days. The solvent was removed on a rotary evaporator (R-220PRO, Büchi do Brazil; Valinhos, Brazil) at 50 °C under reduced pressure, obtaining the crude hydroethanolic extract (CHE, 46.60 g; yield = 11.65%). The CHE (5 g) was solubilized in 70% ethanol (200 mL) and subsequently subjected to liquid–liquid partition in a separating funnel with *n*-hexane, dichloromethane, and ethyl acetate. The solvents were removed using a rotary evaporator (R-220PRO, Büchi do Brazil; Valinhos, Brazil) at 50°C, under reduced pressure, resulting in dichloromethane (DCM, 0.926 g), ethyl acetate (EtOAc, 0.945 g), and hydroethanolic (EtOH, 0.964 g) fractions. The yield of the *n*-hexane fraction was negligable.

### 4.3. Phenolic Compounds Content

Total phenols were quantified using the Folin–Ciocalteau method according to Pastrana-Bonilla et al. [54]. The total phenol content was calculated using a calibration curve with 100 µg mL^−1^ gallic acid solution as the standard. Sample absorbances were read at 760 nm using a UV–visible spectrophotometer (Q798Ul, Quimis; Diadema, Brazil). Determinations were performed in triplicate, and the results are given in microgram equivalents of gallic acid per milligram of dry sample (µg GAEq mg^−1^ DS).

The total flavonoid assay was performed according to Woisky and Salatino [55], and flavonoid content was calculated using a calibration curve with 100 µg mL^−1^ quercetin in a methanol solution of 2% aluminum chloride as a standard. Sample absorbances were read at 425 nm using a UV–visible spectrophotometer (Q798U, Quimis; Diadema, Brazil). Determinations were performed in triplicate, and the results are given in microgram equivalents of quercetin per milligram of dry sample (µg QEq mg^−1^ DS).

### 4.4. HPLC-DAD-MS Analyses 

The chemical composition of the *B. holophylla* crude extract and its fractions were analyzed on an ultra-fast liquid chromatography (UFLC) system (Prominence, Shimadzu; Kyoto, Japan), using an Ascentis Express C18 column (10 cm × 4.6 mm, 2.7 µm (Supelco Analytical; Bellefonte, PA, USA). This was coupled to a diode-array detector (DAD) (SPD-M20A) monitored between 200 and 800 nm, and a quadrupole time-of-flight tandem mass spectrometer (micrOTOF QII, Bruker Daltonics; Massachusetts, USA) equipped with an electrospray ionization (ESI) source. The injection volume was 1 μL of the sample (prepared at a concentration of 1 mg/mL), and the flow rate was 0.3 mL/min. The mobile phase was composed of acetonitrile (solvent B) and deionized water (solvent A) with the addition of 0.1% formic acid (*v*/*v*). The applied elution profile was: 0–5 min: 10 to 20% B, 5–10 min: 20% B, 10–15 min: 20 to 30% B, 15–20 min: 30 to 45% B, 20–33 min: 45 to 100% B, 33–37 min: 100% B, 37–38 min: 100 to 10% B, and 38–42 min: 10% B. Mass spectrometry analysis was applied using the following parameters: capillary voltage, 3.5 kV; *m/z* range, 120–1300; source and desolvation temperature, 220 °C; cone and desolvation gas flow rate, 9.0 L/min. Nitrogen (4 Bar) was used as the collision gas for MS/MS analyses, with collision energies ranging from 25 to 62.5 eV. The high-resolution mass spectrometer was calibrated using a TFA-Na^+^ solution (10 mg/mL). Identification of the compounds in the samples was attained by comparing the retention times, UV, and MS spectra with literature data.

### 4.5. Antibiofilm Formation Assay

#### 4.5.1. Strains and Culture Conditions

Three *Candida albicans* strains, including *C. albicans* SC5314 (ATCC MYA-2876), *C. albicans* ATCC 18804, and *C. albicans* ATCC 10231, were employed in this study. All the strains were originally obtained from the American Type Culture Collection (ATCC). *C. albicans* SC5314 and *C. albicans* ATCC 18804 were kindly provided by Dr. Susana Johann (Microbiology Laboratory, Federal University of Minas Gerais). According to ATCC (2015), *C. albicans* ATCC SC5314 and *C. albicans* ATCC 18804 are sensitive to anidulafungin, voriconazole, itraconazole, fluconazole, micafungin, caspofungin, and 5-flucytosine. *C. albicans* ATCC 10231 is resistant to anidulafungin, voriconazole, itraconazole, fluconazole, and ketoconazole, but is sensitive to micafungin, caspofungin, and 5-flucytosine. The fungal strain cultures were routinely maintained in incubation at 37 °C (SP-101/30, SPLabor; Presidente Prudente, Brazil) in 2% Sabouraud dextrose broth [56].

#### 4.5.2. Microdilution Assay: Effect of the *B. holophylla* Extract and Fractions on *C. albicans* Planktonic Growth

The microdilution method was employed to determine the susceptibility of *C. albicans* strains in planktonic growth according to document M27-A3 of the Clinical and Laboratory Standards Institute, with minor modifications [57]. The *B. holophylla* extract and fractions were freshly dissolved in 2% DMSO at a concentration of 2.5 mg/mL and then two-fold diluted serially to the desired concentrations. In this study, 1 × 10^3^ CFU/mL of the *C. albicans* strains was treated with *B. holophylla* extract and fractions (0.25–1250 µg/mL) by the addition of the respective samples and *C. albicans* suspension in Sabouraud dextrose broth into 96-well round-bottom microplates. Fluconazole was used as a positive control for *C. albicans* SC5314 and *C. albicans* ATCC 18804, and nystatin was used as a positive control for *C. albicans* ATCC 10231. Cells treated with 2% (*v*/*v*) DMSO were used as the vehicle control. Samples under each condition were incubated at 37 °C for 48 h, and the minimum inhibitory concentration (MIC) was determined as the lowest concentration where no visible growth was observed. The assays were performed in triplicate, and the results are expressed as the mean of the three independent experiments.

#### 4.5.3. Antibiofilm Formation Assay

Biofilm formation was determined as described previously by Xu et al. [58], with minor modifications. For the adhesion stage, *C. albicans* strains (100 μL) were seeded at a density of 1 × 10^6^ CFU/mL in 96-well flat-bottom microplates, and incubated at 37 °C for 90 min to allow attachment of the yeast cells on the polystyrene surface. After this, non-adherent cells were removed, and adhered cells were cultured in the presence or absence of *B. holophylla* extract or fractions, freshly dissolved in 2% DMSO at a concentration of 0.25–1250.00 µg/mL. Fresh RPMI 1640 medium (100 µL/well) was added, and the plates were incubated at 37 °C for 48 h. 

To study the effect of *B. holophylla* extract or fractions on mature biofilm, yeast cells were suspended in RPMI 1640 medium (1 × 10^6^ CFU/mL), and 100 µL was transferred into the wells of flat-bottom 96-well plates, and incubated at 37 °C for 48 h to allow biofilm formation and maturation. Next, the biofilms were treated with the extract or fractions (1250 µg/mL) for 48 h. Crystal violet staining (0.1%) was used to measure total biomass [59]. Data are expressed as percentages of biofilm mass in treated samples vs. untreated controls. Nystatin was used as positive control, and RPMI 1640 medium supplemented with 2% (*v*/*v*) DMSO was used as the vehicle control.

#### 4.5.4. Inhibition of Yeast–Hyphae Transition

The effect of *B. holophylla* extract or fractions on yeast–hyphae transition was evaluated according to Andrade et al. [60]. Hyphae growth was induced by incubation of *C. albicans* strains (1 × 10^3^ CFU/mL) in a 96-well microplate with fetal bovine serum and *B. holophylla* extract or fractions (625, 312.50, and 125 µg/mL). The microplates were incubated for 24, 48, and 72 h at 37 °C, and hyphae formation was observed with a light microscope (Primo Star, Zeiss; São Paulo, Brazil) using 400-fold magnification and documented by AxioVision software (Zeiss). The experiments were performed in triplicate, and repeated three times. Positive (nystatin) and solvent (2% *v*/*v* DMSO) controls were included.

### 4.6. Statistical Analysis

All tests were performed in triplicate with at least two independent experiments. The software used for the statistics was GraphPad Prism^®^ v. 8.0.1 (GraphPad Software, Inc. La Jolla; California, USA). An one-way analysis of variance (ANOVA) test was used, followed by a Tukey’s test to compare the results between groups, and Dunnet’s test was used to compare the results between the treatments and the control, with a *p*-value < 0.05 deemed significant. The results are expressed as the mean ± standard deviation.

## 5. Conclusions

The qualitative analysis of the chemical profile of extract and fractions from *B. holophylla* leaves revealed the presence of flavonoids, predominantly of the flavonol-3-*O*-glycoside type based on quercetin, myricetin, and kaempferol. The flavonoid-rich fractions showed a promising effect on biofilm formation and hyphae growth, important virulence factors of *C. albicans* strains. The results confirm the biotechnological potential of *Bauhinia* spp. to produce bioactive compounds of economic and medicinal interest, and suggest that further investigations aimed at the isolation and biological evaluation of isolated compounds are desirable.

## Figures and Tables

**Figure 1 plants-11-01796-f001:**
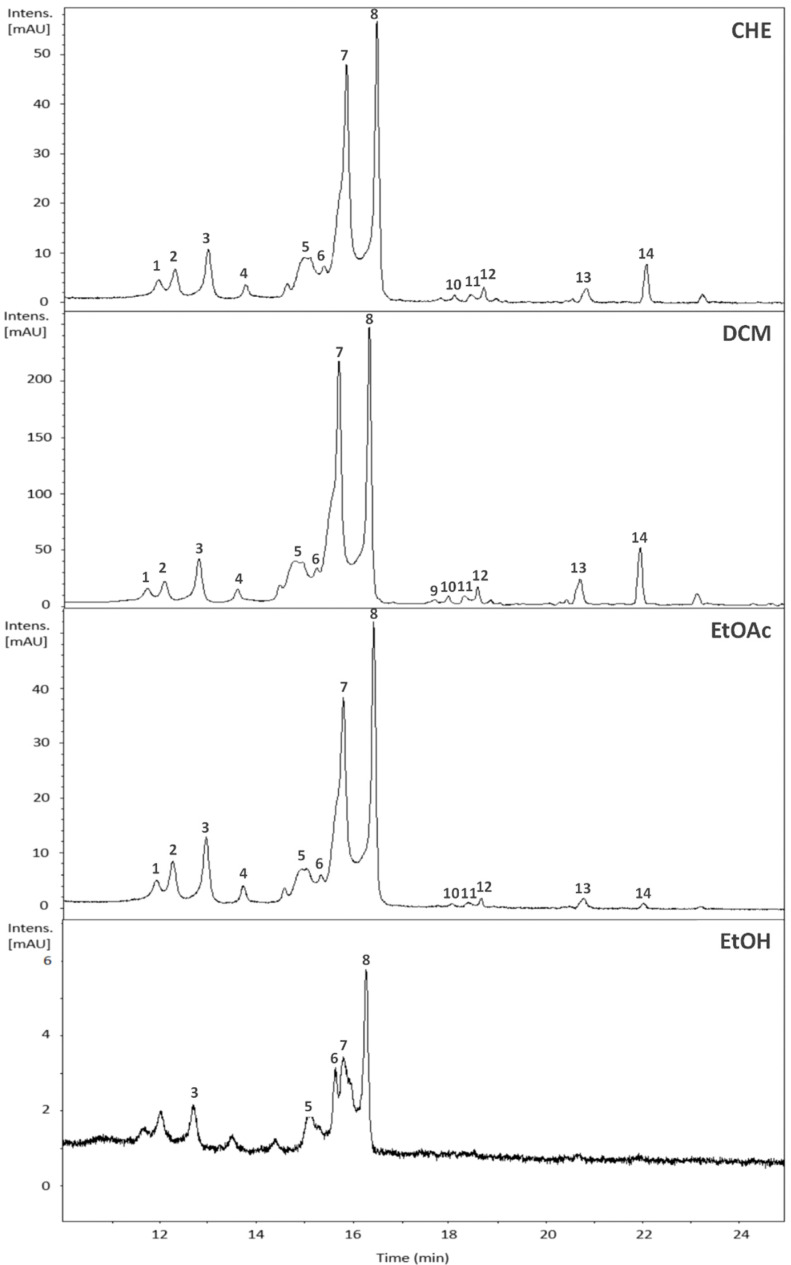
Expansion of total ion chromatograms (10–25 min) of CHE (crude hydroethanolic extract), DCM (dichloromethane fraction), EtOAc (ethyl acetate fraction), and EtOH (hydroethanolic fraction) from *B. holophylla*.

**Figure 2 plants-11-01796-f002:**
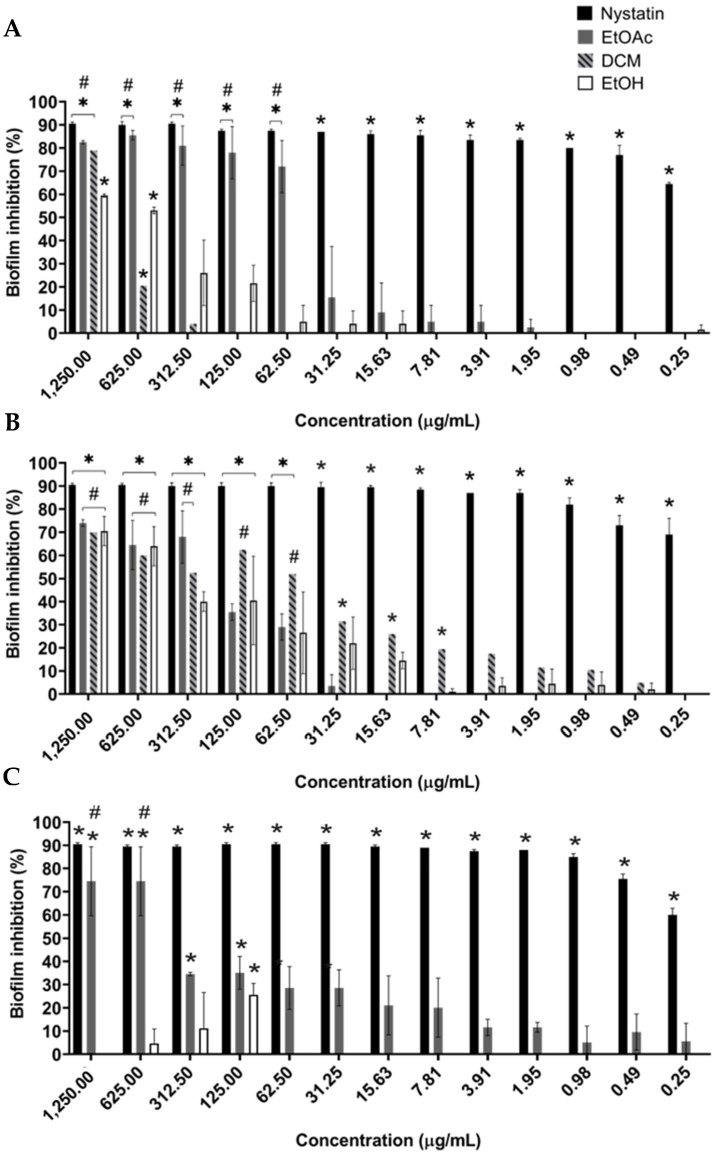
Effect of *Bauhinia holophylla* fractions on biofilm formation of *Candida albicans strains*. (**A**) C. albicans SC5314; (**B**) *C. albicans* ATCC 18804; (**C**) *C. albicans* ATCC 10231. The metabolic activity represented by the *C. albicans* biofilm was compared with the untreated control. The results between groups were compared using Tukey’s test, and Dunnett’s test was used to compare the results between the treatments and the control. Each value is presented as the mean ± standard deviation of two independent experiments (* *p*-value < 0.05). (#) fractions and concentrations do not differ from each other. EtOAc: ethyl acetate fraction; DCM: dichloromethane fraction; EtOH: hydroethanolic fraction.

**Figure 3 plants-11-01796-f003:**
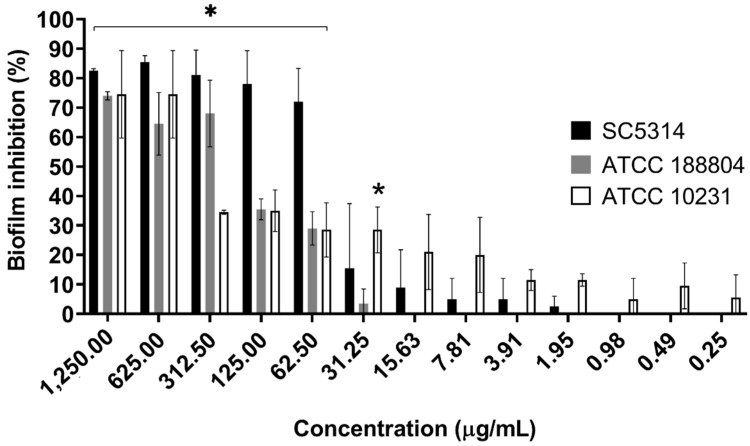
Reduction in biofilm formation of *C. albicans* strains by ethyl acetate fraction. The control represents the culture without any treatment, defined as 0% inhibition. Each concentration had seven replicates. The results between groups were compared using Tukey’s test, and Dunnet’s test was used to compare the results between the treatments and the control. Each value is presented as the mean ± standard deviation of two independent experiments (* *p*-value < 0.05).

**Figure 4 plants-11-01796-f004:**
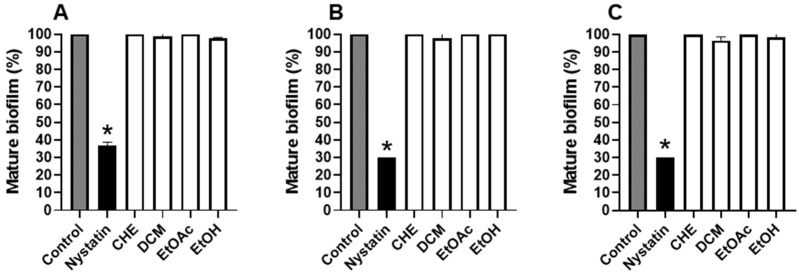
Effects of the *Bauhinia holophylla* extract and fractions (1250 µg/mL) on mature biofilm of *C. albicans*. (**A**) *C. albicans* SC5314; (**B**) *C. albicans* ATCC 18804; (**C**) *C. albicans* ATCC 10231. The control is the absence of treatment and 100% biofilm biomass. The results between groups were compared with Tukey’s test, and Dunnet’s test was used to compare the results between the treatments and the control. The results are presented as the mean ± standard deviation of two independent experiments (* *p*-value < 0.05). CHE: crude hydroethanolic extract; DCM: dichloromethane fraction; EtOAc: ethyl acetate fraction; EtOH: hydroethanolic fraction.

**Figure 5 plants-11-01796-f005:**
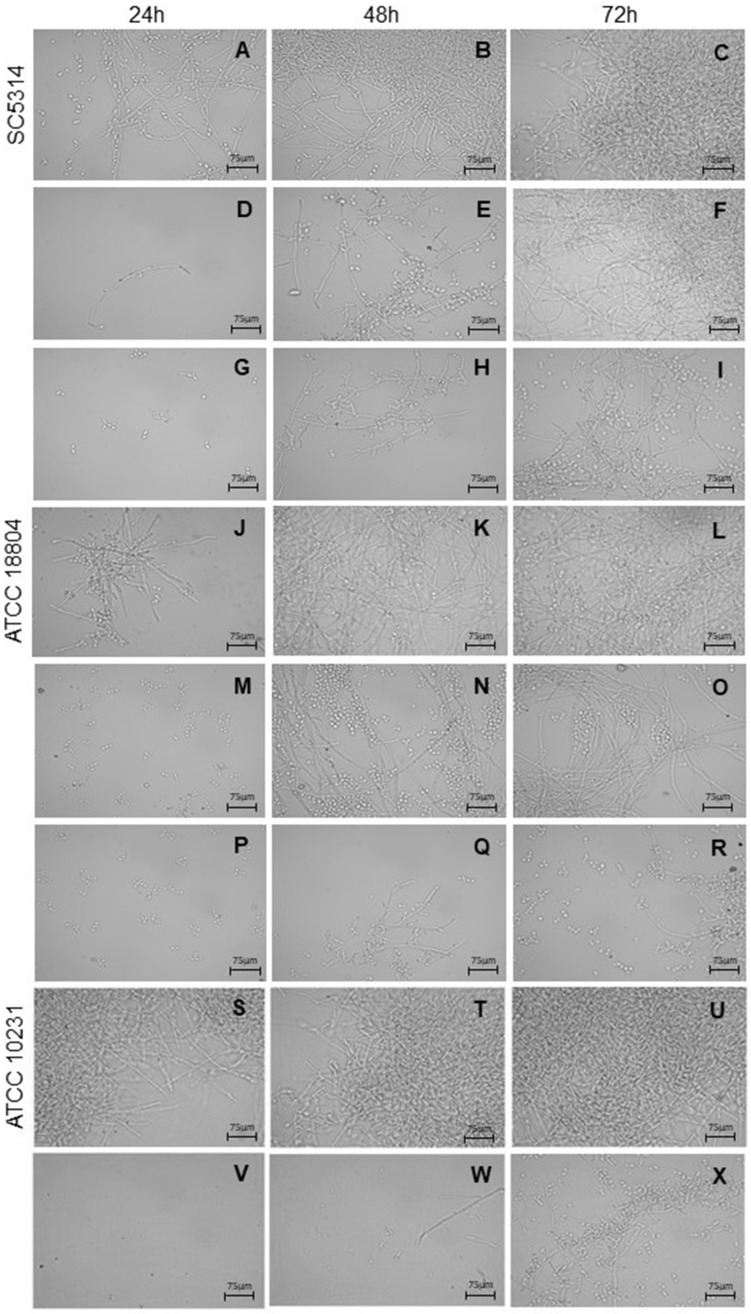
Hyphal formation by *Candida albicans* strains. *C. albicans* was cultured in the absence (control) and presence of *Bauhinia holophylla* extract and fractions at 625, 312.5, and 125 µg/mL for 24, 48, and 72 h at 37 °C. Nystatin was used as a positive control. Only active samples are shown at 625 µg/mL. (**A**–**C**) *C. albicans* SC5314 control; (**D**–**F**) SC5314 with crude hydroethanolic extract; (**G**–**I**) SC5314 with dichloromethane fraction; (**J**–**L**) *C. albicans* ATCC 18804 control; (**M**–**O**) ATCC 18804 with dichloromethane fraction; (**P**–**R**) ATCC 18804 with ethyl acetate fraction; (**S**–**U**) *C. albicans* ATCC 10231 control; (**V**–**X**) ATCC 10231 with hydroethanolic fraction. Representative microphotographs were obtained using 400-fold magnification. Scale bars: 75 µM.

**Table 1 plants-11-01796-t001:** Total phenol and flavonoid contents in the extract and fractions of *B. holophylla* leaves. CHE: crude hydroethanolic extract; DCM: dichloromethane fraction; EtOAc: ethyl acetate fraction; EtOH: hydroethanolic fraction. The results were compared using Tukey’s test, and they are presented as the mean ± standard deviation of three repetitions. Means with the same letter do not differ statistically from each other *(p*-value < 0.05).

Sample	Total Phenols (µg GAEq mg^−1^ DS)	Total Flavonoids (µg QEq mg^−1^ DS)
CHE	39.08 ± 1.77 b	3.91 ± 0.13 c
DCM	37.61 ± 2.54 b	9.93 ± 0.53 a
EtOAc	47.45 ± 1.51 a	6.03 ± 0.35 b
EtOH	20.46 ± 2.32 c	2.37 ± 0.13 c

[Reprinted from: Book, *Phenolic Compounds in Health and Disease*, Phenolic Compounds and Antioxidant and Antibacterial Activities of *Bauhinia holophylla* (Fabaceae: Cercideae), 153–174, Copyright: 2021, Nova Science Publishers, Inc., Authors: Sara Dias da Fonseca, Thaiz Rodrigues Teixeira, Jaqueline Maria Siqueira Ferreira, Luciana Alves Rodrigues dos Santos Lima, João Máximo de Siqueira, Walter Luyten, Ana Hortência Fonseca Castro].

**Table 2 plants-11-01796-t002:** Characterization of compounds in crude extract and fractions from *Bauhinia holophylla* using high-performance liquid chromatography coupled with diode array detector and mass spectrometry (HPLC-DAD-MS) in positive ionization mode. CHE (crude hydroethanolic extract); EtOAc (ethyl acetate fraction); DCM (dichloromethane fraction); EtOH (hydroethanolic fraction). (+) presence and (-) absence. NI = not identified.

Peak	Rt	UVmax	[M + H]^+^ (*m/z*)	[M + H]^+^ (*m/z*)	Error	MS/MS^n^	Compound	Samples				Molecular Formula
	(min.)	(nm)	Experimental	Theoretical	(ppm)			CHE	DCM	EtOAc	EtOH	
1	12.0	254/350	451.0855	451.0871	3.5	319.0412; 303.0472	Myricetin-*O*-pentoside	+	+	+	-	C_20_H_18_O_12_
2	12.3	255/348	465.1013	465.1028	3.2	319.0441; 303.0511	Myricetin-*O*-deoxyhexoside	+	+	+	-	C_21_H_20_O_12_
3	13.0	254/353	465.1004	465.1028	5.2	303.0496	Quercetin-*O*-hexoside	+	+	+	+	C_21_H_20_O_12_
4	13.8	254/350	465.1024	465.1028	0.9	303.0503	Quercetin-*O*-hexoside	+	+	+	-	C_21_H_20_O_12_
5	15.0	255/350	435.0900	435.0922	5.0	303.0493	Quercetin-*O*-xilopyranose	+	+	+	+	C_20_H_18_O_11_
6	15.6	255/349	449.1062	449.1078	3.6	287.0559	Kaempferol-3-*O*-glucoside	+	+	+	+	C_21_H_20_O_11_
7	15.9	255/348	435.0901	435.0922	4.8	303.0499	Quercetin-*O*-pentoside	+	+	+	+	C_20_H_18_O_11_
8	16.5	254/348	449.1056	449.1078	4.9	303.0470	Quercetin-*O*-deoxyhexoside	+	+	+	+	C_21_H_20_O_11_
9	17.7	264/346	419.0959	419.0973	3.3	287.0505	Kaempferol-*O*-pentoside	-	+	-	-	C_20_H_18_O_10_
10	18.1	264/346	479.1168	479.1184	3.3	317.0624; 302.0381	Isorhamnetin-3-*O*-hexoside	+	+	+	-	C_22_H_22_O_12_
11	18.4	264/348	479.0938	-	-	317.0619	Undentified	+	+	+	-	NI
12	18.7	264/345	433.1108	433.1129	4.8	287.0517	Luteolin-deoxyhexose	+	+	+	-	C_21_H_20_O_10_
13	20.8	254/362	303.0493	303.0499	2.0	181.9561	Quercetin	+	+	+	-	C_15_H_10_O_7_
14	22.0	254/356	317.0645	317.0656	3.5	302.0408	Isorhamnetin	+	+	+	-	C_16_H_12_O_7_

**Table 3 plants-11-01796-t003:** Minimal inhibitory concentration (MIC) of extract and fractions of *Bauhinia holophylla* against *Candida albicans* strains. CHE: crude hydroethanolic extract; DCM: dichloromethane fraction; EtOAc: ethyl acetate fraction; EtOH: hydroethanolic fraction; NYS: nystatin; FLU: fluconazole. (-) not tested. The results were compared using Dunnet’s test, and they are presented as the mean of three repetitions. Means with the same letter do not differ statistically from each other (*p*-value < 0.05).

Candida albicans	MIC (µg/mL)				Drugs	
	CHE	DCM	EtOAc	EtOH	FLU	NYS
SC5314	>1250 c	>1250 c	>1250 c	>1250 c	1.95 a	-
ATCC 18804	>1250 c	>1250 c	>1250 c	>1250 c	1.95 a	-
ATCC 10231	>1250 c	>1250 c	>1250 c	>1250 c	-	3.91 b

## Data Availability

The data presented in this study are available within the article.

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
