# Peer review of "Flavonoid-Rich Fractions of Bauhinia holophylla Leaves Inhibit Candida albicans Biofilm Formation and Hyphae Growth"

_plants, 2022, doi:10.3390/plants11141796_

Round 1

Reviewer 1 Report

In this manuscript it was described the effect of extracts and fractions of Bauhinia holophylla on Candida albicans planktonic growth, biofilm formation, mature biofilm and hyphae growth and also it was characterized its flavonoid content. Despite in general the manuscript was well-structured there are some issues that should be improved: 

- line 11, I think that was only evaluated the effect of different extract fractions and not the effect of flavonoids. Please correct. In addition, there are some difficulties in understanding if they considered that evaluated the effects of extracts or fraction of the extracts. 

- line 15, in the definition of abbreviation of HPLC-DAD-MS it was missing the DAD. 

-lines 18-19 and 82, it was not clear if the 3 flavonoids mentioned were the main ones present in the extracts. It was possible to determine de % or concentration of the flavonoids?

- lines 93-104, the effects of the extracts were about 1000 times lower than the positive controls (nystatin and fluconazole). This shows that the effect was very weak compared to the existing drugs meaning that for the obtention of some effects very high concentrations are necessary and consequently toxicity also was expected. Probably some data regarding the toxicity of the extract also should be presented. 

-lines 170 and 190, the authors refer that in figure 4 was presented from V to Y but I did not find the Y image!

-lines 224-227, I think there is no sufficient evidence for this affirmation as it is not described if they are the main constituents of the extract (to claim this information assays with these individual compounds should be performed or at least data from other studies with these strain of C. albicans).

- the conclusions seem to be too much short.

Author Response

Dear Reviewer,

The authors would like to thank you for the review and for your contribution to the improvement of our manuscript. As far as possible, all suggestions and corrections proposed were made.

Please find below the description of all the modifications made:

1) line 11, I think that was only evaluated the effect of different extract fractions and not the effect of flavonoids. Please correct. In addition, there are some difficulties in understanding if they considered that evaluated the effects of extracts or fraction of the extracts.

Authors' response: The requested correction has been made. In the text we consider evaluating the effect of extract and fractions and not the flavonoids.

2) line 15, in the definition of abbreviation of HPLC-DAD-MS it was missing the DAD.

Authors' response: As recommended the definition has been corrected to high-performance liquid chromatography coupled with diode-array detector and mass spectrometry (HPLC-DAD-MS).

3) a) lines 18-19 and 82, it was not clear if the 3 flavonoids mentioned were the main ones present in the extracts.

Authors' response: According to the chemical study of the crude extract and fractions, the samples presented predominance of the flavonoids of the flavonol-3-O-glycoside type based on the aglycones quercetin, myricetin, and kampferol. This information has been corrected in the text.

b) It was possible to determine de % or concentration of the flavonoids?

Authors' response: Since we do not have standards of flavonoids glycosylated with different sugars available in the laboratory, only qualitative and not quantitative analyzes were performed, therefore it is not possible to determine the percentage of each compound in the extract and fractions.

4) lines 93-104, the effects of the extracts were about 1000 times lower than the positive controls (nystatin and fluconazole). This shows that the effect was very weak compared to the existing drugs meaning that for the obtention of some effects very high concentrations are necessary and consequently toxicity also was expected. Probably some data regarding the toxicity of the extract also should be presented. 

Authors' response: Cytotoxicity of extracts and fractions was evaluated against Vero cells by the MTT (3-(4,5-dimethylthiazolyl-2)-2,5-diphenyltetrazolium bromide) colorimetric assay (data not show). The CC50 presented were: crude extract (89.6 µg/mL), ethyl acetate fraction (EtOAc): (93.4 µg/mL), dichloromethane fraction (DCM) (57.0 µg/mL), and hydroethanolic fraction (EtOH) (143.9 µg/mL). However, these data were not showed because we believe that it will be better to present these results in future studies, in which other eukaryotic cells will also be used, since these experiments are still ongoing.

5) lines 170 and 190, the authors refer that in figure 4 was presented from V to Y but I did not find the Y image.

Authors' response: The requested correction has been made. Please, see the Figure 5.

6) lines 224-227, I think there is no sufficient evidence for this affirmation as it is not described if they are the main constituents of the extract (to claim this information assays with these individual compounds should be performed or at least data from other studies with these strain of C. albicans).

Authors' response: The requested correction has been made. The sentence has been rephrased to “In this way, the flavonoid-rich fractions of B. holophylla leaves have potential candidates for use in combination with established antifungal drugs to treat C. albicans infections, since they may act in different stages of biofilm formation”.

7) the conclusions seem to be too much short.

Authors' response: the authors opted for a more direct and succinct conclusion, without repeating the results, but emphasizing the main results, the biotechnological potential of Bauhinia species, and also, the need to isolate the substances present in the fractions, and evaluate the biological activity of the isolated compounds.

Best regards,

The authors.

Reviewer 2 Report

The article is written correctly, its structure is also correct and well thought out and the topic is interesting.

Nevertheless, a few questions raise my doubts:

1) why was only identification performed without quantification of phenolic compounds?

2) if not all identified compounds were quantified, why was Total Flavonoid Content (TFC) not tested?

3) it is necessary to include the chromatogram from the analysis (UV spectrum with which the compounds were identified, i.e. 340/350 or 360 nm or TIC)

4) Table 1 - why are m/z  rounded? It seems to me that with the equipment specified in the methodology at your disposal, you can read values ​​up to 4 decimal places. Please provide m/z measured and theoretical. Please enter the mass error in ppm.

5) Methodology
- please provide all device names in the following format: device (model, company; city, country).
- please provide details of the MS analysis (colision energy, cone voltage, etc.)
- please provide the software used for the statistics

6) Please provide the DOI numbers in Reference section

Author Response

Dear Reviewer,

The authors would like to thank you for the review and for your contribution to the improvement of our manuscript. As far as possible, all suggestions and corrections proposed were made.

Please find below the description of all the modifications made:

1) Why was only identification performed without quantification of phenolic compounds?

Authors' response: Since we do not have standards of flavonoids glycosylated with different sugars available in the laboratory, only qualitative and not quantitative analyzes were performed, therefore it is not possible to determine the percentage of each compound in the extract and fractions.

2) If not all identified compounds were quantified, why was Total Flavonoid Content (TFC) not tested?

Authors' response: The quantification of the phenolic compounds was performed in the crude extract and fractions. However, these data have already been included in another publication. We contacted the publisher that allowed the reproduction of data in the manuscript. The quantification of total phenols and flavonoid is show in the Table 1.

3) It is necessary to include the chromatogram from the analysis (UV spectrum with which the compounds were identified, i.e. 340/350 or 360 nm or TIC).

Authors' response: The chromatogram was included as Figure 1.

4) Table 1 - why are m/z  rounded? It seems to me that with the equipment specified in the methodology at your disposal, you can read values ​​up to 4 decimal places. Please provide m/z measured and theoretical. Please enter the mass error in ppm.

Authors' response: As suggested, the experimental and theoretical m/z data were added to the table with values to 4 decimal places, in addition to the error calculated in ppm.

5) Methodology
- please provide all device names in the following format: device (model, company; city, country).

Authors' response: The requested corrections have been made.

- please provide details of the MS analysis (colision energy, cone voltage, etc.)

Authors' response: The requested details of the MS analysis was provided to the methodology section (4.4. HPLC-DAD-MS analyses).

- please provide the software used for the statistics

 Authors' response: The software was provide in Statistical Analysis -section 4.6.

6) Please provide the DOI numbers in Reference section.

Authors' response: The DOI numbers were provide, except for reference 30 (there is no DOI number for it).

Best regards,

The authors.

Reviewer 3 Report

1.       The extraction method should be very briefly mentioned in the abstract , as well as the fractionation performed.

2.       In the abstract, in lines 19 and 20 it is written “. Flavonoids from B. holophylla leaves displayed 19 antifungal activity and inhibited both biofilm formation and hyphae growth.. ”. This is only partial information since the strains affected are not mentioned in this sentence. You need to review this sentence to include also the names of the affected strains.

3.       In the abstract, in lines 20-21 it is written “This study indicates that flavonoids from B. 21 holophylla leaves interfere with the virulence of Candida species and support the use of Bauhinia spp. 22 in folk medicine to treat infections.”. However, you did not write in the abstract about results of testing for flavonoids, only of extracts.  Either describe results for flavonoids, or refer this results for the extracts and not specifically to flavonoids.

4.       There is a linguistic problem with the sentence in line 52. The logic and structure is flawed. It needs to be corrected.

5.       Statistics: Not all the data are analyzed statistically. This is a major flaw of the manuscript. For example, the data in table 2 is not analyzed statistically. Presenting only averages is insufficient.

6.       Figure 1: the figure is not clear enough. It should be replaced with a figure of higher resolution. Furthermore, the size of the writing in the figure (axes titles, numbers and the legend) are two small and not easily identified. They should be enlarge.

7.       The legend of Figure 1 should mention the type of statistical analyses performed for the analyses of the data.

8.       Figure 2: (same comments as for figure 1): the figure is not clear enough. It should be replaced with a figure of higher resolution. Furthermore, the size of the writing in the figure (axes titles, numbers and the legend) are two small and not easily identified. They should be enlarge. Additionally, the legend of the figure should mention the type of statistical analyses performed for the analyses of the data.

9.       Figure 3: the legend of the figure should mention the type of statistical analyses performed for the analyses of the data.

10.   In line 180, and the word low before the word toxicity in this part of the sentence “Their wide distribution in nature, low cost, and toxicity are”.

11.   In line 220, in the following part of the sentence, replace the word –in- with the word – to-. “facilitate adherence and penetration in the host tissue”.

12.   In section 4.1. Chemicals, as well as throughout the material and methods section and the manuscript, when mentioning in parentheses a manufacturer name and the country, the city as well should be included (Manufacturer, city, country).

13.   Line 300, the sentence should be improved. Specifically, “have are sensitive to..” the word have should be removed.

Author Response

Dear Reviewer,

The authors would like to thank you for the review and for your contribution to the improvement of our manuscript. As far as possible, all suggestions and corrections proposed were made.

Please find below the description of all the modifications made:

1) The extraction method should be very briefly mentioned in the abstract, as well as the fractionation performed.

Authors' response: The requested corrections have been made.

2) In the abstract, in lines 19 and 20 it is written “. Flavonoids from B. holophylla leaves displayed antifungal activity and inhibited both biofilm formation and hyphae growth.. ”. This is only partial information since the strains affected are not mentioned in this sentence. You need to review this sentence to include also the names of the affected strains.

Authors' response: The requested corrections have been made. As all strains were affected by different fractions, in order not to exceed the maximum limit of 200 words in the abstract, we included in the abstract the following information: “in all strains tested”.

3. In the abstract, in lines 20-21 it is written “This study indicates that flavonoids from B. holophylla leaves interfere with the virulence of Candida species and support the use of Bauhinia spp. in folk medicine to treat infections.”. However, you did not write in the abstract about results of testing for flavonoids, only of extracts.  Either describe results for flavonoids, or refer this results for the extracts and not specifically to flavonoids.

Authors' response: The requested corrections have been made. We rephrased to flavonoids-rich fractions as requested by another reviewer.

4) There is a linguistic problem with the sentence in line 52. The logic and structure is flawed. It needs to be corrected.

Authors' response: The requested correction has been made. This sentence was rephrased to “Flavonoids are one of the major group of bioactive compounds found in the Bauhinia genus. They correspond to an important class of plant-derived secondary metabolites...”

5) Statistics: Not all the data are analyzed statistically. This is a major flaw of the manuscript. For example, the data in table 2 is not analyzed statistically. Presenting only averages is insufficient.

Authors' response: According to published papers by our research group, the MIC determination was made in triplicate and repeated three times. We detected a mistake in the caption of the Table 3 “The results were expressed as the mean of three independent experiments”. We changed by “The results were expressed as the value of three independent experiments”. In MIC determination, the mean was not considered, but the value of independente experiments.

6) Figure 1: the figure is not clear enough. It should be replaced with a figure of higher resolution. Furthermore, the size of the writing in the figure (axes titles, numbers and the legend) are two small and not easily identified. They should be enlarge.

Authors' response: The requested corrections have been made. The Figure 1 (now Figure 2) was replaced.

7) The legend of Figure 1 should mention the type of statistical analyses performed for the analyses of the data.

Authors' response: The requested corrections have been made. The information about the statistical analyses was added in the caption of the figure.

8) Figure 2: (same comments as for figure 1): the figure is not clear enough. It should be replaced with a figure of higher resolution. Furthermore, the size of the writing in the figure (axes titles, numbers and the legend) are two small and not easily identified. They should be enlarge. Additionally, the legend of the figure should mention the type of statistical analyses performed for the analyses of the data.

Authors' response: The requested corrections have been made. The Figure 2 (now Figure 3) was replaced and the information about the statistical analyses was added in the caption of the Figure.

9) Figure 3: the legend of the figure should mention the type of statistical analyses performed for the analyses of the data.

Authors' response: The requested corrections have been made. The Figure 3 (now Figure 4)  was replaced to follow the same formatting of the figures 2 and 3. The information about the statistical analyses were added.

10) In line 180, and the word low before the word toxicity in this part of the sentence “Their wide distribution in nature, low cost, and toxicity are”.

Authors' response: The requested correction has been made.

11) In line 220, in the following part of the sentence, replace the word –in- with the word – to-. “facilitate adherence and penetration in the host tissue”.

Authors' response: The requested correction has been made.

12) In section 4.1. Chemicals, as well as throughout the material and methods section and the manuscript, when mentioning in parentheses a manufacturer name and the country, the city as well should be included (Manufacturer, city, country).

Authors' response: The requested correction has been made in section 4.1.

13) Line 300, the sentence should be improved. Specifically, “have are sensitive to..” the word have should be removed.

Authors' response: The requested correction has been made.

Best regards,

The authors.

Reviewer 4 Report

Da Fonseca et al. describe in their manuscript the composition of extracts prepared form Bauhinia holophylla leaves and the antifungal action of these extracts against three Candida albicans strains. The authors conclude that extracts of B. holophylla inhibit the biofilm formation and the hyphae growth of C. albicans, but that these extracts do not efficiently influence the planctonic growth and mature biofilm of these yeast strains.

The whole article is written in a fluid style without any grammar or spelling mistakes. The findings of the phytochemical and the biological part of the manuscript are discussed in detail and altogether 58 references are used. The microbial testing is well performed!

HOWEVER, there some issues as follows which need to be adressed by the authors:

1.         It is excellent to describe the phytochemical composition of the extracts used. The findings are very similar to the phytochemical studies by Camaforte et al., 2019 (see reference No. 25) and Rozza et al., 2015 (see reference No. 27) who reported on the presence of flavonoids. However, leave extracts do not exclusively contain flavonoids, but also other secondary plant products. In the study of Saldanha et al., 2021 (see reference No. 26) cyanoglycosides and also pinitol were elucidated. In the publication by Marena et al., 2021 (see referene No. 30), several classes of natural compounds in B. holophylla leaves are mentioned. In addition publications on other Bauhinia species indicate that leave extracts contain more compounds from other classes of natural products (see e.g. Sebastian et al., 2020, on B. accuminata). Moreover, Fatima et al., 2021, describe in their review on Bauhinia racemosa also coumarins, other phenolic compounds and catechin monomers. Thus the extracts tested in this study are much more complex and contain many more natural compounds which might be important for the effects described by Da Fonseca et al. in this manuscript.

2.        Title: Therefore it is not accurate to use „Flavonoids“ in the title as also other natural products in the extracts might play a role in the antifungals activities. In addition no single flavonoid was tested within this study. Therefore the title in its present form is misleading. The title should therefore be „Flavonoid-rich fractions of Bauhinia holophylla leaves inhibit Candida albicans biofilm formation and hyphae growth“.

3.        Line19: „Flavonoid-rich fractions of B. holophylla leaves displayed …“

4.        Line 21: „This study indicates that flavonoid-rich fractions from B. holophylla …“

5.        Line 196: quercetin, myricetin and kaempferol are aglycones, but not flavonol-3-O-glycosides, therefore the sentence needs to be changed to „… containing flavonol-3-O-glycosides based on the aglyones quercetin, myricetin, and kampferol …“. Please change this also in line 225 and lines 357/358

6.        Please also check throughout the manuscript the issues mentioned in remark 3-5

Other aspects:

7.        Line 69: Please add a reference on the traditional use of Bauhinia species for the treatment of infections!

8.        Lines 76-79: The compound 5 should be „Quercetin-O-xylopyranose“

9.        Lines 76-79: Please explain why compounds 9-12 were not detected in the crude extract, whereas these compounds were present in the DCM extract!

10.     Line 180: I propose to add a „low“ before „toxicity“

11.     Line 300: „have“ should be cancelled

To sum up, the manuscript is written in an excellent style and will be of great interest to the reader and the audience, respectively. Currently there is a great research interest in natural compounds which have antimicrobial activites especially against drug-resistent strains. Therefore this contribution addresses a cutting-edge topic. The findings of Da Fonseca et al. might be of clinical relevance in the future.

Author Response

Dear Reviewer,

The authors would like to thank you for the review and for your contribution to the improvement of our manuscript. As far as possible, all suggestions and corrections proposed were made.

Please find below the description of all the modifications made:

  1. It is excellent to describe the phytochemical composition of the extracts used. The findings are very similar to the phytochemical studies by Camaforte et al., 2019 (see reference No. 25) and Rozza et al., 2015 (see reference No. 27) who reported on the presence of flavonoids. However, leave extracts do not exclusively contain flavonoids, but also other secondary plant products. In the study of Saldanha et al., 2021 (see reference No. 26) cyanoglycosides and also pinitol were elucidated. In the publication by Marena et al., 2021 (see referene No. 30), several classes of natural compounds in B. holophylla leaves are mentioned. In addition publications on other Bauhinia species indicate that leave extracts contain more compounds from other classes of natural products (see e.g. Sebastian et al., 2020, on B. accuminata). Moreover, Fatima et al., 2021, describe in their review on Bauhinia racemosa also coumarins, other phenolic compounds and catechin monomers. Thus the extracts tested in this study are much more complex and contain many more natural compounds which might be important for the effects described by Da Fonseca et al. in this manuscript.

Authors' response: Thank you very much for your comments and considerations.

  1. Title: Therefore it is not accurate to use „Flavonoids“ in the title as also other natural products in the extracts might play a role in the antifungals activities. In addition no single flavonoid was tested within this study. Therefore the title in its present form is misleading. The title should therefore be „Flavonoid-rich fractions of Bauhinia holophylla leaves inhibit Candida albicans biofilm formation and hyphae growth“.

Authors' response: The requested corrections have been made. The title was change as requested.

  1. Line19:“Flavonoid-rich fractions of B. holophylla leaves displayed …“

Authors' response: The requested correction has been made.

  1. Line 21:„This study indicates that flavonoid-rich fractions from B. holophylla …“

Authors' response: The requested correction has been made.

  1. Line 196: quercetin, myricetin and kaempferol are aglycones, but not flavonol-3-O-glycosides, therefore the sentence needs to be changed to „… containing flavonol-3-O-glycosides based on the aglyones quercetin, myricetin, and kampferol …“. Please change this also in line 225 and lines 357/358

Authors' response: The requested correction has been made.

  1. Please also check throughout the manuscript the issues mentioned in remark 3-5.

Authors' response: The requested correction has been made.

Other aspects:

  1. Line 69: Please add a reference on the traditional use of Bauhinia species for the treatment of infections!

Authors' response: The requested correction has been made. The reference was added.

  1. Lines 76-79: The compound 5 should be „Quercetin-O-xylopyranose“

Authors' response: The requested correction has been made.

  1. Lines 76-79: Please explain why compounds 9-12 were not detected in the crude extract, whereas these compounds were present in the DCM extract!

Authors' response: The LC-MS/MS data were analyzed again and it was possible to observe traces of other compounds in the CHE and EtOAc fraction (compounds 10-12) and in addition the compounds 5-7 in the EtOH fraction. In general, some compounds may not have been detected in the crude extract because it is a complex mixture, where signals may overlap or the concentration of these compounds may be very close to or below the detection limit of the equipment, which makes their detection difficult to visualize and consequently non-identification. When the crude extract is subjected to the fractionation process with organic solvents of different polarities, there is a concentration of compounds according to the solubility of the compound in each solvent, so a greater number of compounds was observed, mainly in the DCM fraction.

  1. Line 180: I propose to add a „low“ before „toxicity“

Authors' response: The requested correction has been made.

  1. Line 300: „have“ should be cancelled.

Authors' response: The requested correction has been made.

Best regards,

The authors.

Round 2

Reviewer 1 Report

In general, the comments and suggestions were taken into consideration and therefore the manuscript should be accepted for publication.

Author Response

Dear Reviewer,

The authors would like to thank you for the review and for your contribution to the improvement of our manuscript. 

Best regards.

Reviewer 2 Report

In current form I recommend for publication.

Author Response

(The authors gave the same response as above.)

Reviewer 3 Report

The manuscript was revised sufficiently based on all comments but one: 

Statistical analysis is required for Table 3. You cannot present the data without a statistical analysis. 

Author Response

Dear Reviewer,

The authors would like to thank you for the review and for your contribution to the improvement of our manuscript. Please find below the description of the modification made:

1) Statistical analysis is required for Table 3. You cannot present the data without a statistical analysis. 

Authors' response: Statistical analysis was added in Table 3 as requested by reviewer (please, see lines 134 to 137).

Best regards,

The authors.
